# A diffusion anisotropy descriptor links morphology effects of H-ZSM-5 zeolites to their catalytic cracking performance

Xiaoliang Liu [1,3], Jing Shi[1,3], Guang Yang[1], Jian Zhou[1], Chuanming Wang[1], Jiawei Teng [1✉], Yangdong Wang [1✉] & Zaiku Xie[2✉]

Zeolite morphology is crucial in determining their catalytic activity, selectivity and stability, but quantitative descriptors of such a morphology effect are challenging to define. Here we introduce a descriptor that accounts for the morphology effect in the catalytic performances of H-ZSM-5 zeolite for $C_4$ olefin catalytic cracking. A series of H-ZSM-5 zeolites with similar sheet-like morphology but different $c$-axis lengths were synthesized. We found that the catalytic activity and stability is improved in samples with longer $c$-axis. Combining time-resolved *in-situ* FT-IR spectroscopy with molecular dynamics simulations, we show that the difference in catalytic performance can be attributed to the anisotropy of the intracrystalline diffusive propensity of the olefins in different channels. Our descriptor offers mechanistic insight for the design of highly effective zeolite catalysts for olefin cracking.

[1] State Key Laboratory of Green Chemical Engineering and Industrial Catalysis, Shanghai Research Institute of Petrochemical Technology, SINOPEC Corp., Shanghai, China. [2] China Petrochemical Corporation (SINOPEC Group), Beijing, China. [3]These authors contributed equally: Xiaoliang Liu, Jing Shi. ✉email: tengjw.sshy@sinopec.com; wangyd.sshy@sinopec.com; xzk@sinopec.com

H-ZSM-5 zeolite with MFI topology, has been extensively studied and applied in a series of chemical processes due to its abundant well-defined microporous structures and the intrinsic moderate Brønsted acidities[1,2]. The framework of H-ZSM-5 zeolite contains two types of intersecting 10-membered ring channels[3,4], as shown in Fig. 1a. The straight channels (5.4 × 5.6 Å) run along the crystallographic b-axis with exposed facet of [010] and the sinusoidal channels (5.1 × 5.4 Å) parallel to the a-axis with exposed facets of [101] and [100]. It was reported that regulating the preferred orientations of the pore systems to crystal planes will cause a variety of the H-ZSM-5 morphologies, thus correspondingly affecting the diffusion resistances[5,6]. Morphology adjustment by reducing the particle size and shape with a controllable a/b or a/c "aspect ratios" is considered as an efficient route to reduce the diffusion path lengths, increase the accessibility of active sites, and finally promote catalytic activities of H-ZSM-5 samples[7–12]. In particular, constructing H-ZSM-5 catalysts with intergrowths on well-defined crystal facet opening[7,12] or nanosheet morphology with extremely short thickness along b-axis[8,9,11] shows remarkably longer lifetime in catalytic reactions. Therefore, controlling the morphology of H-ZSM-5 zeolite is of certain considerable importance for the diffusion and shape-selective catalysis, whereas the intrinsic relationship between morphology and catalytic property is obscure due to the lack of diffusive descriptors revealing morphology effect. Practically, it is not always trivial to quantify this relationship and define a suitable descriptor.

To elucidate the key descriptor of morphology effect over H-ZSM-5 zeolite in catalysis process, it is extremely key to differentiate diffusivities in the two types of pore channels[13,14]. The diffusion in the two-channel system of H-ZSM-5 crystal is actually complicated due to the sensitivity to the kinetic diameter of adsorbed molecules and the topological structures of zeolites. The molecules like methanol, xenon, methane, etc. are small enough to diffuse randomly through the network of two intersecting channel systems without configuration limitations[15–17]. However, as the size of guest molecules further increasing, such as short chain hydrocarbons ($C_3$, $C_4$) or aromatic molecules, an anisotropic diffusion would occur, where the escape rates of the adsorbate in the intersecting pores of ZSM-5 zeolites are diverse in different channel segments (Fig. 1b)[9,15,18,19]. Therefore, differentiating diffusivities in straight and sinusoidal channels over H-ZSM-5 zeolites are of benefit to the understanding of activity, stability, or product distribution in catalysis and the guidance of catalyst design with designated morphology. The catalytic cracking of low-value $C_{4+}$ olefin fractions (olefin catalytic cracking (OCC)), deriving from

fluid catalytic cracking and methanol to olefins processes[7,20–22], represents a meaningful route to improve the ethylene and propylene production (important to the polymer industry) using H-ZSM-5 catalysts in industrial application[23–27]. Herein the OCC process was employed to clarify the descriptor of morphology effect in catalysis process by revealing the diffusive behaviors in straight and sinusoidal channels of H-ZSM-5 zeolite.

In this work, several H-ZSM-5 zeolites with controllable sheet-like morphology were designed: various lengths along c-axis with comparative textures, acidities, lengths of a-axis, and thicknesses of b-axis, suggesting equiform diffusion path lengths but different exposed percent of pore channels to defined crystal facets. A homemade time-resolved in situ Fourier transform infrared (FTIR) spectroscopy was developed to study the diffusion properties of zeolite catalysts. Combined with experimental evidences and molecular dynamics (MD) simulations, the relationship between morphologies, catalysis behaviors, and diffusion properties will be discussed in detail. A descriptor about diffusion anisotropy for zeolite's morphology effect in OCC was proposed and integrally investigated. The diffusion anisotropy descriptor was employed to modulate the OCC performances.

## Results

**Structural characterizations of H-ZSM-5 zeolites.** The structural characterizations were performed to gain insights into the structures, chemical properties, and morphologies of the H-ZSM-5 zeolites. A series of designed H-ZSM-5 samples (denoted as Z-cS, Z-cM, and Z-cL) were synthesized and selected for investigation in this work. The X-ray diffraction (XRD) patterns in Supplementary Fig. 1a displays the diffraction peaks of MFI topological structure with considerable crystallographic intensity, indicating a high and similar relative crystallinity over the samples[5,28]. All the Ar physisorption isotherms of the samples show similar shapes (Supplementary Fig. 1b), indicating a comparable pore structure. The pore volumes, Brunauer, Emmett, and Teller (BET) surface areas, and detailed textural parameters over the samples are provided in Supplementary Table 1. There are no obvious distinctions in BET surface areas and pore volumes over the selected samples. The result of inductively coupled plasma analysis indicates that overall samples have nearly similar Si/Al molar ratios with the value of 288, 301, and 297 (Supplementary Table 1h). Brønsted acid sites, as shown at 1545 cm$^{-1}$ in Supplementary Fig. 1c, are deemed to be the active sites for OCC reaction, and their numbers are summarized in Supplementary Table 1i, in which the total amounts of active sites for various samples are comparative.

The scanning electron microscopic (SEM) images of these H-ZSM-5 samples are shown in Fig. 2a, b. All of the samples exhibit a coffin-like shape, which is a typical morphology of MFI-type zeolites[5,28]. The average lengths of a-, b-, and c-axis over the corresponding zeolites were counted from Fig. 2a by statistics of 100 specimens. As shown in Fig. 2a, b and Supplementary Fig. 2, the samples present similar lengths of crystal along the a- and b-axis, with the average size of ~250 and ~100 nm, respectively, whereas the lengths of c-axis are apparently diverse. Z-cS presents the smallest length of c-axis with a medial of 548 nm, while the mean lengths for Z-cM and Z-cL are 970 and 1530 nm, respectively. According to the transmission electron microscopic (TEM) and aberration-corrected scanning transmission electron microscopic (STEM) images, the characteristic 10-membered ring straight channels that run along b-axis can be distinguished along the [010] plane (Fig. 2c–e), and the sinusoidal channels that are parallel to a-axis could be observed along the [100] plane (Fig. 2f–h). Besides, the exposure degree of [010] crystal facets was calculated. The results are summarized in Table 1b, and the

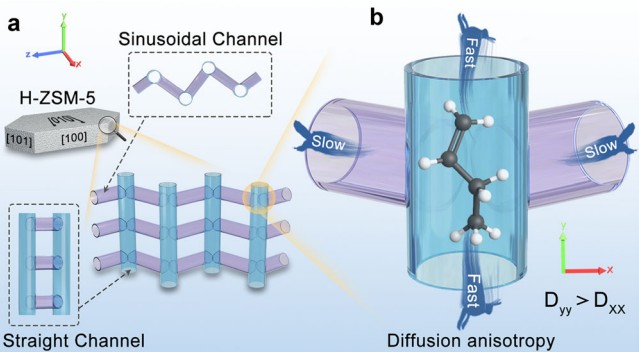

**Fig. 1 The framework and diffusion model of H-ZSM-5-type zeolites composed of two-channel directions. a** The straight and sinusoidal channels running along b- and a-axis with similar opening sizes but different geometry shapes. **b** Schematic view of diffusion anisotropy in two-channel network of H-ZSM-5 zeolite.

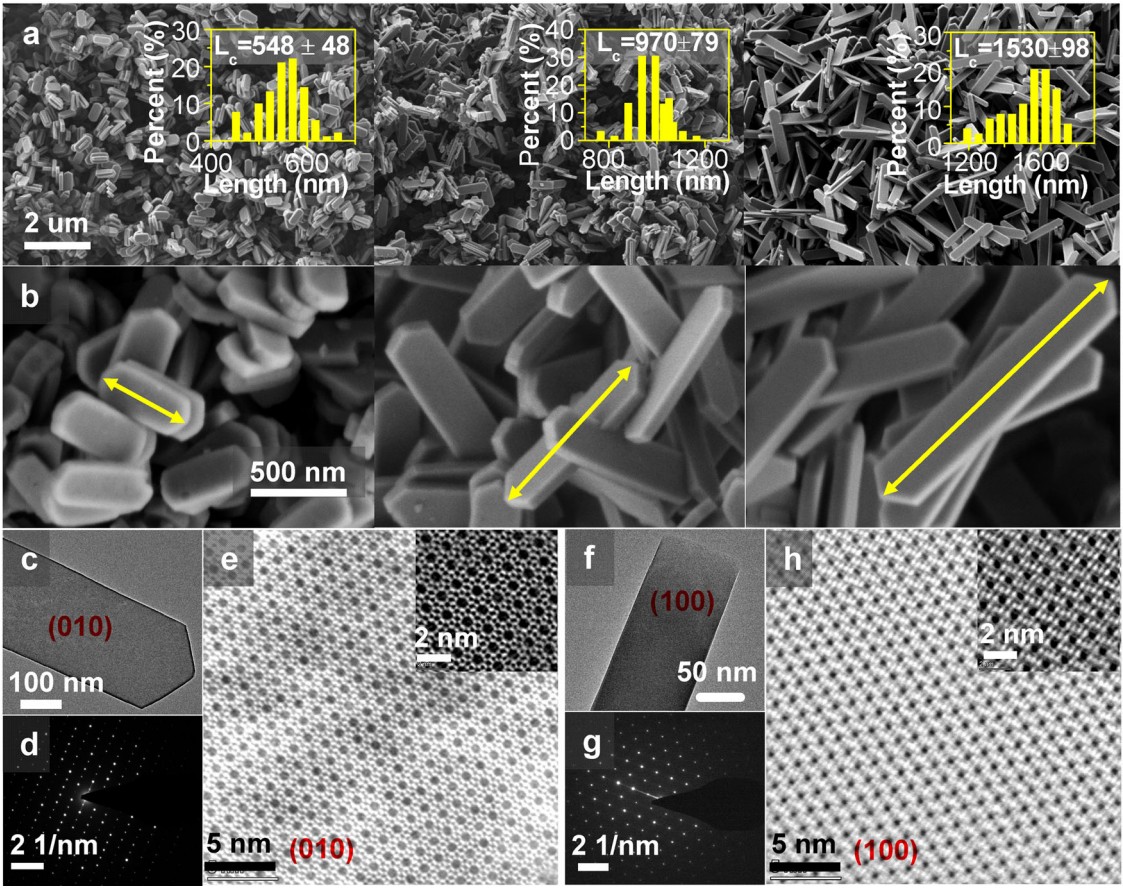

**Fig. 2 The morphology of the as-synthesized H-ZSM-5 samples. a, b** SEM results of H-ZSM-5 zeolites with similar thicknesses of *a*- and *b*-axis but different lengths of *c*-axis. Left: Z-cS; middle: Z-cM; right: Z-cL. **c** TEM image of Z-cL sample with exposed facets of [010] plane. **d** The corresponding SAED patterns of [010] plane. **e** Aberration-corrected STEM image of Z-cL sample with exposed facets of [010] plane, with the inset on the top right displaying the STEM image with high resolution and the framework structure of H-ZSM-5 projected along [010] plane. **f–h** TEM image, the corresponding SAED patterns, and the aberration-corrected STEM image of Z-cL sample with exposed facets of [100] plane.

**Table 1 Morphology parameters of the as-synthesized H-ZSM-5 samples.**

| Samples | $L_a$[a] [nm] | $L_b$[a] [nm] | $L_c$[a] [nm] | [010] exposure degree[b] [%] | Straight channel percent[c] [%] | Sinusoidal channel percent[d] [%] |
|---------|--------|--------|--------|-------------------|---------------------|----------------------|
| Z-cS | 248 | 100 | 548 | 63.0 | 67.6 | 32.4 |
| Z-cM | 246 | 97 | 970 | 67.2 | 69.7 | 30.3 |
| Z-cL | 249 | 98 | 1530 | 68.9 | 70.4 | 29.6 |

The $n_{[010]}$, $n_{[100]}$, and $n_{[101]}$ are used to represent the pore amounts per $nm^2$ over corresponding crystal facet. Detailed calculations for the surface areas of exposure facets are displayed in the "Methods" section (Eqs. 3–5).
[a]Determined by statistics of 100 specimens in SEM images displayed in Fig. 2.
[b]Calculated by the exposure surface areas of various crystal planes. $d_{[010]} = S_{[010]}/(S_{[010]} + S_{[100]} + S_{[101]})$.
[c]Calculated by the following expression: $p(str.) = S_{[010]} \times n_{[010]}/(S_{[010]} \times n_{[010]} + S_{[100]} \times n_{[100]} + S_{[101]} \times n_{[101]})$.
[d]Calculated by the following expression: $p(sin.) = 100\% - p(str.)$.

detailed calculation procedures are displayed in the "Methods" section. All of these sheet-like zeolites exhibit a dominant exposed plane of [010]. Longer *c*-axis results in a higher exposed degree of [010] plane and Z-cL presents the highest exposed degree up to 68.9%.

Furthermore, the lattice parameters for the H-ZSM-5 crystal unit cell have been decided from XRD methodology, and the diameters of *a*-, *b*-, and *c*-directions in one unit cell are considered as 20.07, 19.92, and 13.42 Å, respectively, in the *Pnma* space group (orthorhombic), as shown in the databases of IZA structure. The same results were reported in some early papers[15,16,18]. Therefore, the cross-section parameters of a unit cell in various crystal facets could be presented. The [010] plane

in one unit cell exhibits two pores follow the straight channel path, with a cross-section of 20.07 × 13.42 Å. The [100] and [101] direction in one unit cell possess two pores follow the sinusoidal channel path, with cross-sections of 19.92 × 13.42 Å and 19.92 × 24.14 Å, respectively. Thus, the degree of exposed pore channels per unit area ($n^{-2}$) in corresponding exposed plane can be calculated, as listed in Supplementary Table 2. Then the proportions of the two types of channels in the selected H-ZSM-5 samples can be explicitly computed, as presented in Table 1c, d, and the detailed calculation processes are displayed in Table 1 and the "Methods" section. It can be found that, as the exposed degree of [010] plane increased, the more percent of straight channels are revealed.

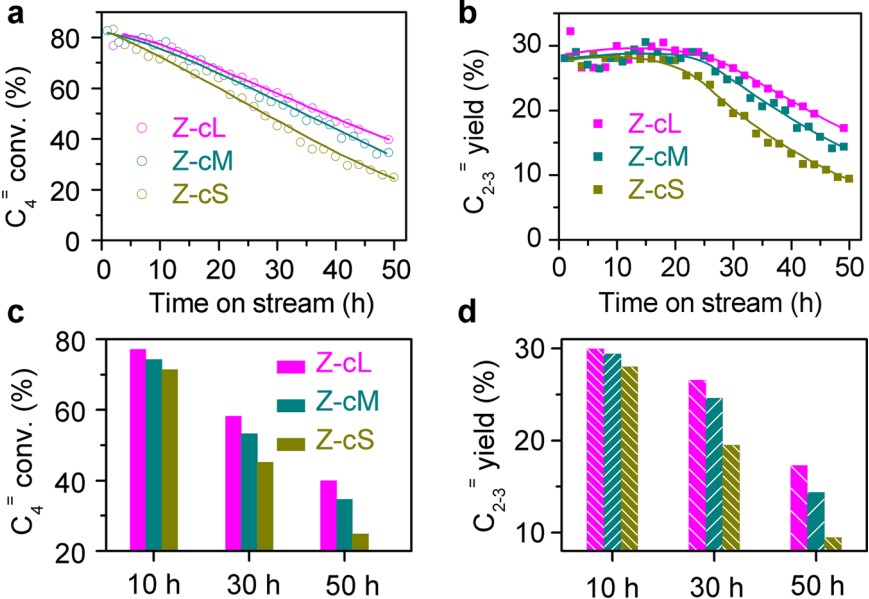

**Fig. 3 Catalytic behaviors for $C_4$ olefin catalytic cracking reactions over Z-cS, Z-cM, and Z-cL catalysts. a** $C_4^=$ conversions. **b** Yields of propene and ethene. **c**, **d** detailed $C_4^=$ conversions and $C_{2-3}^=$ yields at different reaction times. Reaction conditions: $M$ (catalyst) = 0.3 g; $P$ = 1.6 bar; $T$ = 823 K; $F$ = 15 mL h$^{-1}$; time on stream, 50 h.

**Catalytic behaviors over H-ZSM-5 catalysts with various morphologies.** In this work, the catalytic behaviors on the cracking of $C_4$ olefin (OCC) were explored over H-ZSM-5 catalysts with various sheet-like morphology, and the results are displayed in Fig. 3. The OCC reactions were performed with a high weight hourly space velocity of pure butene (30 h$^{-1}$) at 823 K, as shown in Fig. 3a, b. In the initial few hours, all of these catalysts presented a relatively stable performance; the $C_4^=$ conversion and yield of the main products $C_{2-3}^=$ over Z-cL are slightly higher than these over the other two catalysts. Further prolonging the reaction time would obviously vary the catalytic performances, and the activities are decreased in the sequence of Z-cL > Z-cM > Zc-S. To directly find out the explicit catalytic distinctions over the catalysts, the detailed $C_4^=$ conversions and $C_{2-3}^=$ yields at various reaction times are displayed in Fig. 3c, d. At initial 10-h reactions, $C_4^=$ conversions are kept >70%, with lower olefins yields of almost 30% over all catalysts. The activity of Z-cL is slightly higher than that of the other two catalysts. After reacting for 50 h, Z-cL presents a much slow decreasing tendency, of which $C_4^=$ conversion and $C_{2-3}^=$ yield could retain 40 and 18%, respectively, which is higher than that of the other two catalysts. Generally, the reduction of particle size would help to ensure stability regarding catalyst deactivation for long-term catalytic reactions[8–10]. However, our experimental facts in Fig. 3 point out that enlarging the size along c-axis in H-ZSM-5 catalyst can result in a higher OCC catalytic activity and a better stability. It is well known that many factors in H-ZSM-5 crystal can influence the catalysis properties, such as acidities[26,29], textural structures[27], and morphology[9–12]. On the one hand, the influence of the textural structures and acid sites in our disquisitive catalysts can be excluded due to the similarity in these zeolites, as presented in Supplementary Fig. 1 and Supplementary Table 1. On the other hand, the characters of morphology are significantly different over those selected catalysts, as shown in Fig. 2 and Table 1, which could be attributed to the probable influencing factor for the catalytic performance.

**Diffusion behaviors of morphology effect over H-ZSM-5 catalysts.** To obtain better cognition of the relationship between the catalytic properties and the morphology, the uptake rates of $C_4$ molecules (the kinetic diameter is comparable to the pore sizes of H-ZSM-5) over those three samples were compared using our homemade time-resolved in situ FTIR spectroscopy. Detailed exploratory operation conditions about this measurement as well as the original uptake curves are shown in Supplementary Figs. 3 and 4. The diffusion rate ($D_{eff}/L^2$) was applied to evaluate the diffusion resistance of these analytic samples, which can be fitted from the normalized uptake curves with Eq. 6 in the "Methods" section. Lower $D_{eff}/L^2$ represents a stronger diffusion resistance in zeolite pore systems. Figure 4a–c displays the normalized uptake curves of $C_4$ molecules adsorption over various zeolites, and the uptake rates are increased in the order of Z-cS < Z-cM < Z-cL, indicating that the longer c-axis morphology in H-ZSM-5 is leading to the faster diffusion rate. Figure 4d presents the correlation of the diffusivity with the exposure degree of [010] plane over these analytic H-ZSM-5 samples. The diffusivity increases almost linearly with the exposure degree of [010] plane over H-ZSM-5 samples. Meanwhile, the uptake properties of the main product molecule ($C_3^=$) were performed over the identical samples. As displayed in Supplementary Fig. 5, the same results for $C_3^=$ molecule were obtained as that of $C_4^=$ molecule that enlarging the exposure degree of [010] plane lead to a faster diffusion rate.

Moreover, the diffusion behaviors of Z-cS and Z-cL catalysts for butene molecules were performed with a temperature range of 323–423 K (Fig. 4e, f). Here a lower temperature range was applied for the diffusion measurements since 473 K is high enough to cause the polymerization of $C_4^=$ molecule (Supplementary Fig. 6). As displayed in Fig. 4g, the apparent diffusive rates over Z-cS are lower than Z-cL at all of the temperatures (detailed fitted data are listed in Supplementary Table 3), and the activation energy of Z-cS (26.4 kJ mol$^{-1}$) is 30% higher than that of Z-cL (20.3 kJ mol$^{-1}$). These results further demonstrate that the increase of [010] exposure degree in H-ZSM-5 crystal can reduce the diffusion resistance, resulting in a lower active energy barrier to a certain extent.

The results presented in Fig. 4 have quantificationally indicated that the exposed degree of crystal facet orientation is responsible

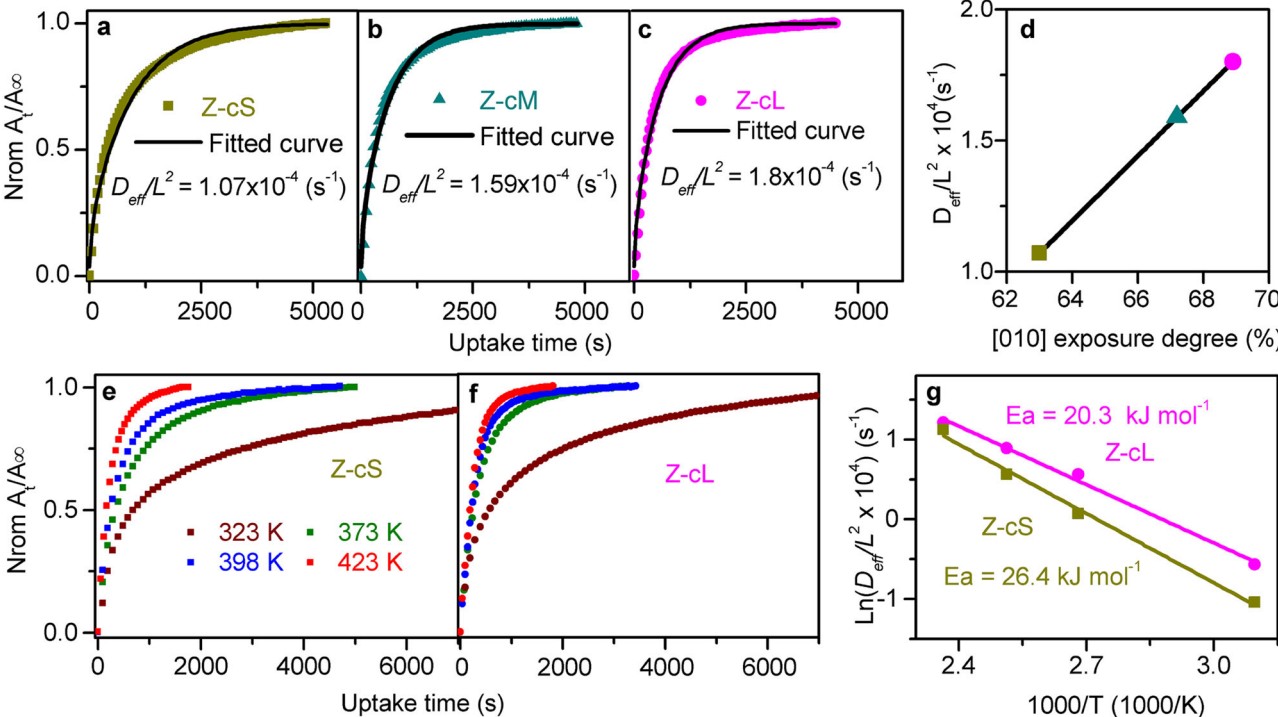

**Fig. 4 Diffusion behaviors of $C_4^=$ in H-ZSM-5 zeolites.** Uptake curves of $C_4^=$ over H-ZSM-5 samples with different length of *c*-axis: **a** Z-cS, **b** Z-cM, **c** Z-cL. **d** Correlation of the [010] exposure degrees of H-ZSM-5 zeolites with the diffusion rates fitted by Eq. 6. **e**, **f** Uptake curves of $C_4^=$ over Z-cS and Z-cL with different temperature. **g** Arrhenius plot of diffusion rates over Z-cS and Z-cL. The detailed conditions of diffusive measurements are displayed in the "Methods" section. The error bars are smaller than the data points in Fig. 4d, g.

for the various diffusion barriers in H-ZSM-5 crystals. It is worth noting that the diffusivity derived from the time-resolved in situ FTIR spectroscopy is actually the apparent diffusive rates reflecting the combination of surface and intracrystalline diffusion in the nanosheet H-ZSM-5 crystals[30,31]. The surface diffusive mechanism, though remaining unrevealed, is closely related to the external surface characteristics of H-ZSM-5 crystals[31,32]. The intracrystalline diffusion is directly attributed to the properties of nanoporous intracrystalline pore systems[33,34]. Recently, Gao et al.[35,36] have proposed an approach to directly quantify dual resistance model (DRM) by deducing an approximate expression relying solely on surface permeability from the initial uptake rate ($\alpha/L$):

$$\frac{M_t}{M_\infty}\bigg|^{\sqrt{t}\to 0} \cong \frac{\alpha}{L}(\sqrt{t})^2 + O(\sqrt{t^3}) \qquad (1)$$

And the intracrystalline diffusivity can be fitted by the following approximation[35,37,38]:

$$\frac{L^2}{D_{eff}} = \frac{L^2}{D_{intra}} + \frac{3L}{\alpha} \qquad (2)$$

where $D_{eff}$ represents the apparent diffusion coefficient, $D_{intra}$ is the intracrystalline diffusive coefficient, $\alpha$ is the surface permeability, and $L$ is the diffusion path length.

In order to reveal the intrinsic diffusive mechanism of the morphology effect, initial normalized uptake curves presented in Fig. 3a–c were fitted with Eqs. 1 and 2. The fitted curves and the corresponding results of $D_{intra}/L^2$ and $\alpha/L$ are displayed in Fig. 5a–d, respectively. As the enhancement of exposed degree of the [010] plane, similar $\alpha/L$ and apparent increase in $D_{intra}/L^2$ can be observed, which means that the properties of intracrystalline channels instead of the external surfaces are the decisive factors for the diverse diffusion behaviors of morphology effect. The structural characterizations summarized in Table 1 have demonstrated that

the intracrystalline diffusion paths of these three samples are very close due to their similar lengths of *a*- and *b*-axis, which are parallel to the diffusion channels, but the percent of straight channels in H-ZSM-5 increases with the enhancement of the exposure degree of [010] plane. Thus, the percent of straight channels were correlated with the intracrystalline diffusivity ($D_{intra}/L^2$) over various H-ZSM-5 samples, and the results are presented in Fig. 5e. The intracrystalline diffusivity accelerates with the increasing of the percent of straight channels. When the intracrystalline diffusivity was correlated with the percent of sinusoidal channels, an opposite result can be revealed, as shown in Supplementary Fig. 7. Therefore, it can be considered that the diffusion rate in straight channels is faster than that in sinusoidal channels for olefin molecules, which can explain the reason why the percent of straight channels accelerates the intracrystalline diffusivity significantly.

According to the results presented above, it can be seen that the diffusion barriers over H-ZSM-5 zeolites are significantly determined by controlling the morphology with various exposed facet degrees after excluding the influence of the textural compositions and active sites. The diffusion anisotropy in intracrystalline channels of H-ZSM-5 crystal is considered to be the essential factor for morphology effect. Due to the preferred diffusive pathway, increasing the straight channel percent by enlarging the [010] plane degree is an efficient route to accelerate the internal diffusive rate.

MD simulations were further carried out to investigate the anisotropic diffusion of guest molecules in two channels of MFI-structured zeolite. A total of 32 propene or 24 1-butene molecules were loaded in the supercell ($2 \times 2 \times 2$) of MFI-structured zeolite. As shown in Fig. 6, the mean square displacements (MSDs) for each molecule in *x*, *y*, and *z* directions were derived and linear relations with time are held at the temperatures of 323, 373 and 423 K. It can be seen that the time dependence of MSDs varies with the diffusion directions. It is clear that the diffusion of

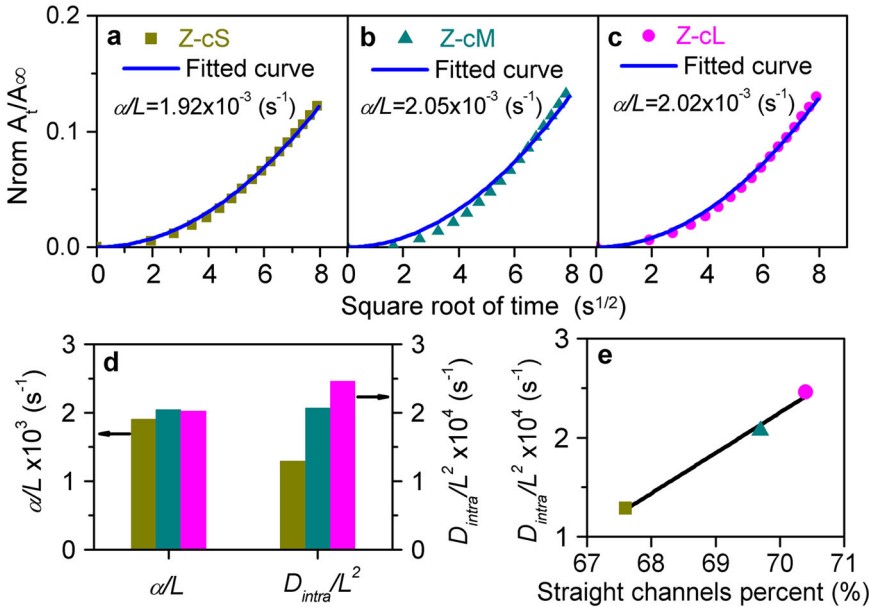

**Fig. 5 The intracrystalline and surface diffusion behaviors of $C_4^=$ molecule in H-ZSM-5 zeolites.** Initial uptake rates of butene molecule over H-ZSM-5 zeolites: **a** Z-cS, **b** Z-cM, and **c** Z-cL. **d** Surface permeability ($\alpha/L$) and intracrystalline diffusion rates ($D_{intra}/L^2$) of butene molecule derived by the uptake rates of H-ZSM-5 samples (as presented in Figs. 3a and 4a–c) following Eq. 2. **e** Correlation of straight channel percent computed in Table 1 with the intracrystalline diffusion rates ($D_{intra}/L^2$) of butene molecule. The error bars are smaller than the data points in Fig. 5e.

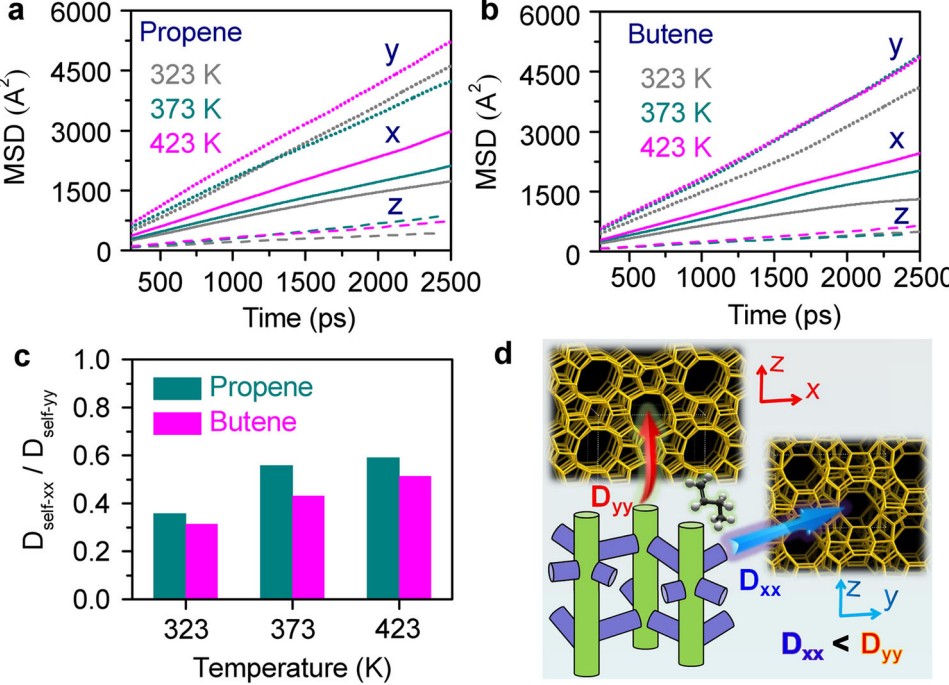

**Fig. 6 Molecular dynamics simulations on the diffusion of propene and 1-butene molecules in MFI-structured zeolite. a** MSDs of propene diffusion in different directions at different temperatures. **b** MSDs of 1-butene diffusion in different directions at different temperatures. **c** The ratio of self-diffusion coefficient in x- and y-directions. **d** Schematic view of olefin molecules diffusion along x- and y-directions of MFI-type zeolites. The anisotropic components $MSD_{xx}$, $MSD_{yy}$, and $MSD_{zz}$ curves were calculated by three individual samples, and the average MSD, $D_{self-xx}$, and $D_{self-yy}$ are reported in this work.

propene and 1-butene along the straight channel (b-direction) are faster than that along the sinusoidal channels (a- and c-directions). The diffusion coefficients along x- and y-directions ($D_{self-xx}$ and $D_{self-yy}$) were then obtained from the slopes of these curves at different temperature (Eqs. 10 and 11 described in the "Methods" section), as listed in Supplementary Tables 4 and 5. Herein the ratio of $D_{self-xx}/D_{self-yy}$ was employed to describe the

diffusion anisotropy of guest molecules in the straight and sinusoidal channels (see Fig. 6c). It is seen that these ratios for propene and 1-butene diffusion increase monotonously with the temperature and the values are <0.6, indicating that the contribution of the sinusoidal channels for the diffusion of both olefins could be enhanced with temperature. In addition, the contribution the straight channels for the diffusion of higher olefins increases as

the ratio for propene is higher than that of 1-butene. These simulation results clearly demonstrate the diffusion anisotropy of olefins in MFI-structured zeolite, which depends on the temperature of diffusion and structure of guest molecules.

**Catalysis–diffusion relationship.** The understanding of the relationship between the catalysis and the diffusion behavior is crucial to future H-ZSM-5 catalyst design for industrial catalysis. The experimental facts in Fig. 3 have pointed out a result that enlarging the particle size along $c$-axis over H-ZSM-5 catalyst can prolong the catalytic lifetime in OCC reaction. It was revealed that the intracrystalline diffusive rates of reactants over those selected catalysts are different due to the diverse characters of two-channel network, as shown in Figs. 4 and 5, which could lead to the differences of catalytic properties. Therefore, the catalytic properties, expressed by the deactivation rates of $C_4^=$ conversion and $C_{2-3}^=$ yield, were correlated with the intracrystalline diffusive rates of reactant molecules in internal pore channels over H-ZSM-5 catalysts, as displayed in Fig. 7a. It can be seen that the deactivation rate distinctly slowed down with the increasing of $D_{intra}/L^2$, indicating that the intracrystalline diffusivity plays a pivotal role in the catalytic cracking reaction over H-ZSM-5 zeolite. For the lower olefin selectivity, as shown in Supplementary Fig. 8, Z-cL with faster intracrystalline diffusivity exhibits higher $C_{2-3}^=$ selectivity, while Z-cS with stronger internal diffusion resistance shows a lower product selectivity.

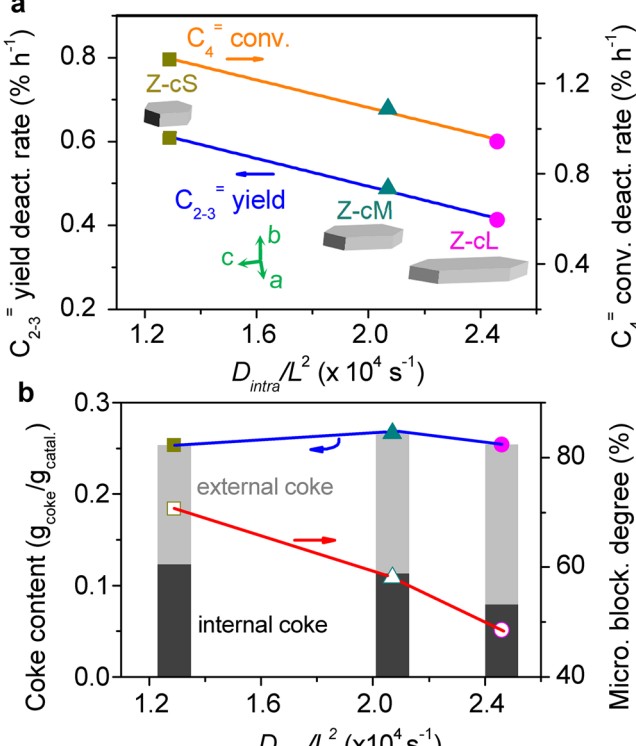

**Fig. 7 The relationships between the intracrystalline diffusivity with the OCC activity losses and the coke depositions. a** Correlation of $C_4^=$ conversions and $C_{2-3}^=$ yield deactivation rates with the intracrystalline diffusive rates of butene molecules in H-ZSM-5 crystals. The deactivation rates were calculated on the basis of the catalytic performances displayed in Fig. 3a, b. The data between 20 and 50 h were chosen due to the actual deactivation time. **b** The coke depositions and the micropore blockage degrees after 50 h of OCC reactions. The error bars are smaller than the data points.

In order to further investigate how intracrystalline diffusivity is related to the catalytic lifetime and product selectivity, the three H-ZSM-5 catalysts were comparatively analyzed using thermogravimetric analysis (TGA) and Ar adsorption after 50 h of reaction (Supplementary Fig. 9 and Supplementary Table 6). Particularly, the locations and the amounts of coke formation calculated using the method reported by Ryoo et al.[39] were studied as a function of the intracrystalline diffusivity. As the result in Fig. 7b shows, although the total coke contents are similar among these three catalysts, it is of interest that the ratios of external and internal coke species are quite different. It was suggested that the coke species deposited inside the micropores is believed to induce the quick deactivation in catalysis[39,40]. The acceleration of $D_{intra}/L^2$ increased the fraction of external coke species rather than that of internal cokes inside the micropores and decreased the microporous blockage degree, hence contributing to prolonging the catalytic lifetime. Thus, a faster intracrystalline diffusive rate would promote the reactants and products to move outward the crystals during OCC reaction and delay further transformation of light olefins to heavy cokes in the micropores, which increases the resistance to deactivation and lower olefin selectivity.

Recently, a descriptor-based methodology has been established by correlating the catalytic properties with a few well-defined properties of a material such as adsorption energies in metal catalysis[41–44]. This methodology enabled the future designing of a highly efficient metal catalyst. For zeolite catalysis, this descriptor-based methodology is still incipiently and principally aims at focusing exclusively on acid sites currently[45]. Because of the similarities of the acid sites over these analytic H-ZSM-5 zeolites (Supplementary Table 1), the reaction pathway of OCC process would predominantly originate from the diffusional effects related to the zeolite morphology. However, it is not always trivial to quantify the relationship between the morphology effects with the catalysis behaviors and define a suitable descriptor in H-ZSM-5 zeolite catalysis. It has been indicated that the intracrystalline diffusion in H-ZSM-5 crystal, which significantly impacts the catalytic performance, is closely related to the percent of straight channels due to their molecule diffusive propensity. Based on the descriptor-based methodology, we consider that this anisotropic diffusion behavior in two-channel network of H-ZSM-5 crystal is a suitable descriptor for morphology effect in OCC. The controlling of the percent of straight channels by prolonging the $c$-axis morphology in H-ZSM-5 could be applied to modulate catalyst lifetime and product yield in OCC.

It should be noted that the enlargement along "$c$" direction over Z-cL zeolite led to not only a higher exposed degree of straight pore channels but also an incremental number of channel intersections. It is well known that the higher proportion of intersections would create a larger extent of intermediate states, which will either remain at the intersections participating in cokes formation leading to fast deactivation or crack to smaller molecules form via $\beta$-scission events in narrow pore channels. Subsequently, the activities or selectivity in OCC process will be affected[46]. In this work, based on the results of the less activity loss and internal coke deposition over Z-cL zeolite, as shown in Fig. 7, we speculate that, although part of the intermediate species remains at these incremental intersections leading to coke precursors, the major fractions are able to crack to smaller molecule form via $\beta$-scission events due to their faster intracrystalline diffusivity of Z-cL catalyst. However, the direct quantification of the effect degrees of these incremental channel intersections on the performance of OCC is actually not revealed in this work, and evaluating these effects is the subject of future work.

## Discussion

In this work, a series of H-ZSM-5 zeolites with controllable sheet-like morphology were designed: the acidities, the textures, the sizes along $a$- and $b$-axis are comparative, but the lengths of $c$-axis are variable, suggesting that the compositions and the diffusive path lengths over these zeolites are constant, but the percents of pore channels are diverse. Then the catalytic performances of these H-ZSM-5 zeolites in OCC were investigated. It was pointed out that enlarging the particle size along $c$-axis in H-ZSM-5 catalyst will improve the catalytic activity and stability. Furthermore, based on the time-resolved in situ FTIR spectroscopy, It was found that the apparent diffusive rates of the guest molecules were significantly boosted with the increase of exposed [010] plane degrees over H-ZSM-5 zeolites. According to the analysis of the DRM, it was demonstrated that the morphology effect of diffusive properties was essentially related to the anisotropic diffusion in different channels of H-ZSM-5 zeolite. The intra-crystalline diffusive rate in straight channels is faster than that in sinusoidal channels for olefins molecules, which was further confirmed by MD simulations. Therefore, the diffusion anisotropy in different channels was proposed as the descriptor for the morphology effect in OCC. The controlling of the intracrystalline diffusive rate via changing the proportion of pore channels could modulate the catalytic activity and stability due to the differences in the location of the coke species in OCC reaction. In short, this work not only provides a clear diffusion anisotropy descriptor to reveal the morphology effect but also offers an insight into the design of highly effective zeolite catalysts for OCC process.

## Methods

**Synthesis of H-ZSM-5 catalysts**. The zeolite materials were prepared by hydrothermal synthesis according to previously reported conventional method[11]. The synthesis of H-ZSM-5 zeolites was carried out by dissolving $Al_2(SO_4)_3 \cdot 18H_2O$ in the solution of sodium hydroxide, followed by the addition of silica solution under stirring. Zeolite synthesis using additives was performed at an original gel composition at 6 $TPA^+$:20 $SiO_2$:2.5 $Na_2O$:0.025 $Al_2O_3$:(0-1) pyrocatechol:80 EtOH:800 $H_2O$, under basic solution with the help of Free Slate High-throughput equipment. The precursor was aged for 24 h and heated to 433 K under 150 rpm stirring for 2 days. After washing and filtering for several times, the products were collected, centrifuged, filtered, and dried in air in the oven. Finally, the samples were calcinated at 823 K for 5 h to remove the organic template.

**Characterizations of H-ZSM-5 catalysts**. XRD patterns were recorded on a Bruker D8 Advance diffractometer. The Cu Kα radiation ($\lambda = 5.1540589$ nm) generated at 40 kV and 40 mA was used as the X-ray source with the scanning rate 1° $min^{-1}$ over a 2θ range of 5–50°. Ar physisorption measurements were performed on a Micromeritics TriStar3000 Surface Area Analyzer at 75 K. Prior to the measurements, the sample was degassed at 623 K until a stable vacuum of 0.67 Pa was reached. The acidity of the H-ZSM-5 samples (Brönsted acid sites) were studied by pyridine adsorption in an in situ IR cell with $CaF_2$ windows. Self-supported sample discs with weight of 15 mg and diameter 13 mm were pretreated at 673 K for 4 h under ultrahigh vacuum system. Subsequently, the sample was cooled down to 473 K, and pyridine steams were admitted to the IR cell for 30 min. Then physical adsorption pyridine was eliminated by evacuation of the zeolite sample for 30 min, and the spectra were recorded on a Nicolet 6700 instrument equipped with an MCT detector at a resolution of 4 $cm^{-1}$. Field emission SEM analysis was performed on a Hitachi S4800 electron microscope. The sample was dispersed ultrasonically in ethanol for 10 min and then was dropped onto a silicon pellet, followed by drying for 60 min. More than 100 particles were used to evaluate the mean length of $a$-, $b$-, and $c$-axis from SEM images. TEM images were performed on a Tecnai 20 STWIN electron microscope at 200 kV. The aberration-corrected STEM images were obtained on a FEI Titan Cubed Themis G2 300 kV with an accelerating voltage of 300 kV.

**Coke analysis**. The H-ZSM-5 zeolite after 50 h of reaction was utilized for TGA (SDT Q600 V20.9 Build 20). The temperature was increased to 1123 K under flowing air (100 mL $min^{-1}$) at a constant ramping rate of 10 K $min^{-1}$. The weight loss between 673 and 1073 K was regarded as the total coke content. The coke density, regarded as similar to that of coal, was estimated to be 1.22 g $cm^{-3}$ [39,47]. On the one hand, the content of internal coke species in the micropores was calculated by the micropore volume decrease, as compared with the fresh zeolite micropore volume. On the other hand, The content of external coke species outside the external surface was determined from the total coke content subtracting the

internal coke content. The micropore blockage degree was obtained by the ratio of micropore volume decrease to the fresh H-ZSM-5 micropore volume.

**Calculations of the surface areas of various exposure facets**. For the coffin-like H-ZSM-5 crystal, the surface areas of [010], [100], and [101] crystal planes were calculated by the following formula:

$$S_{[100]} = 2 \times L_b \times \left( L_c - 2 \times \frac{L_a}{2 \times \tan(118°/2)} \right) \tag{3}$$

$$S_{[010]} = 2 \times L_a \times \left( L_c - \frac{L_a}{2 \times \tan(118°/2)} \right) \tag{4}$$

$$S_{[101]} = 4 \times L_b \times \frac{L_a}{2 \times \sin(118°/2)} \tag{5}$$

where $L_a$, $L_b$, and $L_c$ represents the length of $a$-, $b$-, and $c$-axis in Table 1. 118° represents the angle between two intersecting [101] crystal plane in coffin-like H-ZSM-5 crystal.

**Catalytic cracking process**. The catalytic cracking of butene reactions were performed in a stainless catalytic reactor. The catalyst (0.30 g) with grain sizes of 20–40 mesh was loaded in the reactor and was pretreated in high-purity $N_2$ flow at 823 K for 2 h. Then pure $C_4$ olefin with WHSV of $30^{-1}$ h was introduced into the reactor with a laboratory-scale piston pump for $C_4$ olefin input. The pressure of reactant was typically regulated at 1.6 bar. Products were analyzed by an online gas chromatography (Agilent 7860). The calculation of $C_4^=$ conversion was based on the moles of $C_4^=$ at inlet and outlet gases; the product selectivity was calculated on a molar carbon basis.

**Measurements of diffusivity**. The diffusion properties of guest molecules over as-synthesized H-ZSM-5 samples were evaluated by a homemade time-resolved in situ FTIR spectroscopy. Karge and Niessen first exploited this technique to study the diffusion behaviors of guest molecules over zeolites in 1991[48]; the counter- or co-diffusion behaviors of hydrocarbons in zeolites could be determined under the rigorous conditions of high temperatures and pressures[48–51]. The IR spectra were collected on a Nicolet 6700 instrument equipped with an MCT detector at a resolution of 4 $cm^{-1}$. First, the H-ZSM-5 sample was pressed into a self-supporting wafer with a diameter of 13 mm at the pressed pressure of 30 MPa and placed inside a gold sample holder surrounded by a heating wire. The sample holder was fixed in the in situ IR cell connected to an air control system. Before measurement, H-ZSM-5 sample was pretreated with a 100 mL $min^{-1}$ pure $N_2$ flow at 673 K for 2 h. Subsequently, the sample was cooled down to the test temperature with a 800 mL $min^{-1}$ $N_2$ flow rate; after stabilizing at the targeted temperature, the background spectrum was obtained as a reference in pure $N_2$ flow. For a typical measurement, the guest molecule ($C_4^=$ or $C_3^=$) mixing with inert $N_2$ (0.1 vol% guest molecule + 99.9 vol% $N_2$) with a partial pressure of 1 mbar was introduced into the IR cell with a 800 mL $min^{-1}$ flow, and the IR spectra were collected at 0.96-s intervals. The normalized areas of the IR bands of butene and propene molecules at 2750–3150 $cm^{-1}$ were used to quantify the relative concentrations of adsorbate in the zeolites. Finally, the original and normalized uptake curves were obtained, and the diffusion rates were calculated using the following formula for sheet-like zeolite[52,53]:

$$\frac{m_t}{m_{(t \to \infty)}} = 1 - \frac{8}{\pi^2} \sum_{n=0}^{\infty} \frac{1}{(2n+1)^2} \exp\left( -(2n+1)^2 \frac{\pi^2 D_{\text{eff}}}{L^2} t \right) \tag{6}$$

where $m_t$ is the amount adsorbed at time $t$ (in s), $m_\infty$ is the amount adsorbed at equilibrium coverage, $D_{\text{eff}}$ represents the apparent diffusion coefficient, and $L$ represents the diffusion length of H-ZSM-5 crystal. In this work, the diffusion rate, $D_{\text{eff}}/L^2$ was chosen for evaluating the diffusion property of H-ZSM-5 sample.

**Computational methods**. MD simulations by Forcite module in materials studio 8.0 with COMPASS-II force field were performed to investigate the diffusion properties in two-channel network over MFI-type zeolite. Guest molecule loadings in pure silica zeolite were set at 32 and 24 molecules per supercell ($2 \times 2 \times 2$) for propene and butene molecules, respectively, ensuring that the number of heavy atoms in each simulation was consistent. The initial structural model for each MD simulation was obtained by the Packing technology in the Amorphous Cell module. All MD simulations were performed in NVT ensemble and the temperatures were maintained by the Nose thermostat. The long-range interaction was calculated by Ewald summation method with a cutoff radius 15.5 Å. The velocity Verlet integrator is used with a time step of 1 fs. Snapshots of the positions were recorded at every 0.5 ps. The simulation studies were performed at various temperatures of 323, 373, and 423 K, each for 5 ns, following an equilibration of 0.3 ns. According to the dynamic trajectory of each system, the MSD of an adsorbate molecule during a time interval $\tau$ was calculated by the following equation:

$$\text{MSD}(\tau) = \frac{1}{N_m} \sum_i^{N_m} \frac{1}{N_i} \sum_{t_0}^{N_i} \left| r_i(t_0 + \tau) - r_i(t_0) \right|^2 \tag{7}$$

where $N_m$ corresponds to the number of olefin molecules considered in the calculation of the MSD. Therefore, the self-diffusion coefficient $D_{self}$ was obtained by fitting the MSD plots with respect to the time range 0.3–2.5 ns.

$$MSD(\tau) = 6D_{self}\tau + b \qquad (8)$$

where $b$ is the offset at time zero. In addition to the isotropic average, the anisotropic components of MSD (i.e., $xx$, $yy$, $zz$) were also produced with the following equation:

$$MSD = MSD_{xx} + MSD_{yy} + MSD_{zz} \qquad (9)$$

The self-diffusion coefficient $D_{self\text{-}xx}$ and $D_{self\text{-}yy}$ in the $x$- (sinusoidal channels of MFI-type zeolite) and $y$- (straight channels of MFI-type zeolite) directions could be, respectively, obtained by fitting the separate plots $MSD_{xx}$ and $MSD_{yy}$:

$$MSD_{xx}(\tau) = 2D_{self-xx}\tau + b \qquad (10)$$

$$MSD_{yy}(\tau) = 2D_{self-yy}\tau + b \qquad (11)$$

## Data availability
The data supporting the findings of this study are available from the corresponding author Z.X. on reasonable request.

## Code availability
The code used in this paper is available from the corresponding author Z.X. upon reasonable request.

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

## Acknowledgements

We greatly acknowledge the financial support from the Chinese Postdoctoral Science Foundation (2020M681445), the National Natural Science Foundation of China (92034302), and the Shanghai Rising-Star Program (21QB1406500; 18QB1404500). We would also like to greatly thank Dr. Hao Tian, Dr. Xiejun Huang, and Dr. Jinshu Tian for reviewing the manuscript and offering very helpful suggestions.

## Author contributions

X.L. performed most of the experiments and analyzed the experimental data and also co-wrote the paper. J.T. and Y.W. analyzed the data and co-wrote the paper. J.S. synthesized most of the zeolite catalysts. G.Y. and C.W. performed MD simulations. J.Z. analyzed the results, performed the experiments, and reviewed the paper. Z.X. designed the study, analyzed the experimental results, and co-wrote the paper. All authors discussed the results and analyzed the manuscript.

## Competing interests

The authors declare no competing interests.
