## [Peer Review File · Communications Chemistry]

This manuscript has been previously reviewed at another Nature Research journal. This document only contains reviewer comments and rebuttal letters for versions considered at Communications Chemistry.

REVIEWERS' COMMENTS:

Reviewer #1 (Remarks to the Author):

The work enclosed in this manuscript (MS) introduces a descriptor related to the morphology effect of ZSM-5 zeolites when used as catalysts for the catalytic cracking of C4 olefins. The manuscript is well written; the results are interesting, coherent and are clearly exposed and illustrated, and the work is suitable for publication in Commun Chem, now that the authors have addressed all the suggestions from the first revision.

I only have one comment on the responses to my previous remarks (see comment 7 in the Response to Reviewer document). I agree with the authors that acidity is comparable for all samples and should, therefore, have no influence on the catalytic performance, and that all zeolites studied are ZSM-5 with the same MFI structure, so the dimensions of the pores or of the channel intersections are the same. However, when increasing the size of the crystals along the "c" direction, the number of channel intersections also increases, and this will affect the selectivity of the OCC process, as pointed out by Sarazen et al. (Ref. 46). or by del Campo et al. (Ref. 7).

Response to Reviewer

Response to Reviewer 1

General comment: The work enclosed in this manuscript (MS) introduces a descriptor related to the morphology effect of ZSM-5 zeolites when used as catalysts for the catalytic cracking of C4 olefins. The manuscript is well written; the results are interesting, coherent and are clearly exposed and illustrated, and the work is suitable for publication in Commun Chem, now that the authors have addressed all the suggestions from the first revision.

Reply: We sincerely appreciate the positive evaluation and valuable comment from the reviewer again. We have added additional discussions about the effect of the intersection channels for OCC process in the revised manuscript. Our reply to the comment is described as follow.

Comment: I only have one comment on the responses to my previous remarks (see comment 7 in the Response to Reviewer document). I agree with the authors that acidity is comparable for all samples and should, therefore, have no influence on the catalytic performance, and that all zeolites studied are ZSM-5 with the same MFI structure, so the dimensions of the pores or of the channel intersections are the same. However, when increasing the size of the crystals along the “c” direction, the number of channel intersections also increases, and this will affect the selectivity of the OCC process, as pointed out by Sarazen etl al. (Ref. 46). or by del Campo et al. (Ref. 7).

Reply: We thank the reviewer for this kind comment. After a careful consideration, we agree with the point that the increased numbers of channel intersections in Z-cL catalyst would affect the reaction pathway of catalytic cracking process. we have added the following descriptions about “*the catalysis-diffusion relationship*” in this revised manuscript: “*It should be noted that the enlargement along “c” direction over Z-cL zeolite led to not only a higher exposed degree of straight pore channels, but also an incremental number of channel intersections. It is well known that the higher proportion of intersections would create a larger extent of intermediate states, which*

*will either remain at the intersections participating in cokes formation leading to fast deactivation, or crack to smaller molecules form via β -scission events in narrow pore channels. Subsequently, the activities or selectivity in OCC process will be affected⁴⁶. In this work, based on the results of the less activity loss and internal cokes deposition over Z-cL zeolite, as shown in Fig. 7, we speculate that although part of the intermediate species remains at these incremental intersections leading to coke precursors, the major fractions are able to crack to smaller molecules form via β -scission events due to their faster intracrystalline diffusivity of Z-cL catalyst. However, the direct quantification of the effect degrees of these incremental channel intersections on the performance of olefin catalytic cracking is actually not revealed in this work, and evaluating these effects is the subject of future work. ” **(Please see the last paragraph of “the catalysis-diffusion relationship” section at Page 14.)***